# Off-Road Construction and Agricultural Equipment Electrification: Review, Challenges, and Opportunities

**Fuad Un-Noor** [1,*], **Guoyuan Wu** [1], **Harikishan Perugu** [2], **Sonya Collier** [3], **Seungju Yoon** [3], **Mathew Barth** [1] **and Kanok Boriboonsomsin** [1]

[1] College of Engineering—Center for Environmental Technology and Research, University of California, Riverside, 1084 Columbia Avenue, Riverside, CA 92507, USA

[2] California Department of Transportation, 703 B St, Marysville, CA 95901, USA

[3] Research Division, California Air Resources Board, 1001 I Street, Sacramento, CA 95814, USA

[*] Correspondence: aunn001@ucr.edu; Tel.: +1-9512318334

**Abstract:** Though the current wave of electric vehicles is transforming the on-road passenger and commercial vehicle fleets, similar attempts in the off-road equipment sector appear to be lacking. Because of the diverse equipment categories and varied applications, electrifying off-road equipment requires significant research and development. A successful electrification of such equipment can offer an array of benefits, including reduced air and noise pollution, higher energy efficiency, and increased productivity. This paper provides a review of the current state of technology in off-road equipment electrification, with a focus on the equipment used in construction and agricultural applications. The paper also discusses advantages of, and challenges associated with, electrifying off-road construction and agricultural equipment. In addition, potential solutions for overcoming these challenges as well as opportunities to facilitate the electrification of off-road construction and agricultural equipment are identified.

**Keywords:** agricultural equipment; construction equipment; electric vehicle; hybrid electric vehicle

## 1. Introduction

Electric vehicles (EVs) have become a symbol for emissions reduction in the on-road transportation sector. The superior torque and lower emissions of these vehicles as well as other advantages they offer have generated significant interest in them [1]. Though passenger EVs have faced challenges of limited driving range and insufficient charging infrastructure, those have been gradually overcome. To date, much work has been conducted on the electrification of on-road vehicles [1–3]—from light-duty passenger cars and sports utility vehicles (SUVs) to medium- and heavy-duty commercial trucks—many of which have become commercially available [4–7]. However, pieces of off-road equipment such as those used in construction and agricultural applications have not been given the same level of attention.

Construction and agricultural equipment can be a major source of air pollution in many areas [8], and electrifying a large number of them holds significant promise not only in improving local air quality but also for reducing fuel use and greenhouse gas emissions [9]. Increasingly stringent regulations aimed at reducing these emissions are one of the primary reasons that prompted off-road equipment manufacturers to explore electrification [10]. In the United States, off-road diesel engines over 50 horsepower (hp)/37 kW were first brought under federal emission standards in 1994. These very first standards were called the Tier 1 standards. Tier 1 was subsequently succeeded by Tiers 2, 3, and 4; each more stringent than the former. These emission standards essentially dictate the amounts (gram/kWh) of emissions such as carbon monoxide (CO), nonmethane hydrocarbon (NHMC), oxides nitrogen (NOx), and particulate matter (PM) allowed for different engine power levels. Up until Tier 3, advanced engine designs along with some use of

exhaust gas aftertreatment were adopted to meet these emission limits. Tier 4 standards were introduced in 2004, requiring almost a 90% reduction in NOx and PM emissions. Manufacturers implemented control techniques such as advanced exhaust aftertreatment in the equipment they produced to attain that goal. Tier 5 was presented to the public in 2021 to further reduce NOx and PM emissions [11]. Emission standards at the European Union (EU) followed a similar pattern. The first of EU legislations, the Stage I/II regulations for off-road equipment, were put into effect in 1997. Stage III/IV were introduced in 2005. Then came Stage V, made effective for engines above 130 kW and below 56 kW from 2019; the engines within the 56–130 kW range were brought under Stage V restrictions from 2020. Similar to the United States Tiers, the EU Stages also dictated permissible emission quantities (gram/kWh) for off-road diesel engine exhaust gases—CO, hydrocarbon (HC), NOx, and PM—for different engine power levels [12]. With ever-tightening emission limits, manufacturers have been pushed to develop increasingly advanced engine technologies. Now, as the limits have started stretching the limits of diesel engine technology, different electrification approaches such as mild hybridization and battery electrification has been picked up by manufacturers to comply with the future emission regulations. The state of California has even stated a goal to make all heavy-duty vehicles zero-emission by 2045 for feasible operational cases [13]. Off-road equipment is a major portion of these heady-duty vehicles, and thus it is now essential to start developing electric heavy-duty equipment to have working products on the market by 2045.

To date, much of the electrification efforts for off-road construction and agricultural equipment has focused primarily on the use of diesel-electric and hybrid powertrains, although there have been efforts towards battery electrification. However, due to the unique usage and working conditions of off-road equipment, the electrification technologies used in on-road vehicles may not be directly transferable to off-road equipment [14,15]. For example, hybrid systems from on-road EVs are not directly applicable to hybrid excavators because of the dissimilar working environments [16]. Moreover, the components of off-road electric equipment have to withstand a higher level of impact and vibration compared to those of on-road EVs. For example, the power electronics must be capable of withstanding elements such as mud and water, and the hydrogen tanks of off-road fuel cell electric vehicles (FCEV) must be rugged enough to maintain integrity upon impact.

Previously, research on off-road equipment electrification has been conducted on some specific equipment types (e.g., excavators and tractors) or components (e.g., drivetrain and energy storage system (ESS)), as summarized in Table 1. However, there is an absence of comprehensive review on the state of technology of off-road equipment electrification.

**Table 1.** Examples of previous works on off-road construction and agricultural equipment electrification.

| Reference | Year | Topic |
|---|---|---|
| Yang et al. [17] | 2009 | • Analysis of emission from transportation sector in California and their mitigation |
| Parsons et al. [18] | 2014 | • Off-road drivetrain and battery technologies |
| Aydin et al. [19] | 2014 | • Permanent magnet synchronous generators for off-highway heavy-duty series hybrid application |
| Wang et al. [16] | 2017 | • Hybrid excavators developed by different organizations<br>• ESS configurations and control strategies<br>• Energy savings and challenges of different ESS configurations |
| Kwon et al. [20] | 2010 | • Hybrid excavators employing supercapacitors |
| Wang et al. [21] | 2009 | • Powertrain and performance analysis of hybrid hydraulic excavators |

**Table 1.** *Cont.*

| Reference | Year | Topic |
|-----------|------|-------|
| Zhang et al. [22] | 2019 | • Configurations and energy management strategies of hybrid construction equipment |
| Moreda et al. [23] | 2016 | • Electrification of agricultural tractors |

This paper is aimed at bridging that gap by reviewing the electrification of off-road equipment in construction and agricultural sectors. There are a variety of equipment types and sizes in these two sectors [24], but this paper is focused primarily on those with power ratings of 75 horsepower or more. Off-road equipment generally employs power takeoff (PTO), which is the process of driving accessories using power from the engine. Electrification of PTO is also included in this review as it can result in less use, or more efficient use of the internal combustion engine (ICE), which would reduce emissions [25]. This claim was supported by Wagh et al. [26] who pointed out that alongside the drivetrain, accessories as well as safety and control features could be electrified to provide notable benefits. Along with the configurations of electric off-road equipment presented in previous works, this paper also reviews the energy recovery techniques employed. In addition, the advantages of electric off-road equipment, their technical and operational challenges, and potential solutions are discussed. Lastly, opportunities to facilitate the electrification of off-road construction and agricultural equipment are identified.

The rest of the paper is organized as follows. Section 2 describes the vehicle configurations presented in previous works. Section 3 studies the energy recovery techniques for construction and agricultural equipment. The advantages and challenges of off-road equipment electrification are discussed in Section 4, along with potential solutions. The outcomes of this study and future research topics are delineated in Section 6. Finally, the conclusions are drawn in Section 7.

## 2. Electric Powertrain Architectures in Different Off-Road Equipment Categories

While many publications focus on specific equipment types (e.g., tractors, excavators) or categories (e.g., construction, agriculture), several others are geared toward general, multi-purpose off-road equipment. This section provides an overview of notable works conducted on construction and agricultural equipment electrification, with the additional inclusion of some general off-road equipment types. Each subsection covers different hybrid and battery electric powertrain configurations. A hybrid electric vehicle (HEV) uses electric motor(s) alongside an ICE, while a battery electric vehicle (BEV) employs electric motor(s) exclusively. A separate classification worth mentioning is fuel cell electric vehicle (FCEV), which uses fuel cells to generate electricity for running its electric powertrain. Partial or full electrification of equipment attachments, which conventionally is powered by ICE through PTO, is also discussed in this section. In addition, the maturity level of technology development—software-based simulation or hardware implementation (either on test bench or in vehicle)—is also noted.

### 2.1. General Off-Road Equipment

A variety of EV architectures can be applied to electrify construction and agricultural equipment. Zhang et al. [27] showed the design of a battery management system (BMS) [28] for a light-duty off-road parallel plug-in hybrid (PHEV) vehicle, where they employed fuzzy programming to accomplish the task. Parsons et al. [18] showed the design of a heavy military vehicle employing a series hybrid configuration with hub-mounted electric motors utilizing a two-speed transmission. They stated that the design is scalable to vehicles requiring an individual motor capacity up to 400 kW, so it might be possible to adopt this design for heavy construction equipment. A concept similar to that proposed in Parsons et al. [18] was previously presented by Jackson et al. [29], and a two-speed transmission was also used for hybrid heavy off-road machinery by Sinkko et al. [30].

With sufficiently mature battery technology, the ICE might be discarded to move towards the BEV architecture, especially in applications where duty cycles do not demand power exceeding the battery capacity. Baronti et al. [31] proposed a BMS for lithium iron phosphate (LiFePO$_4$) batteries intended for off-road BEV usage, considering battery modules with four cells. Their goal was to design a system that did not require any bespoke hardware, and could serve a wider range of applications. Employing hydrogen fuel cells to power an electric drivetrain represents another possibility for electrifying off-road construction and agricultural equipment. It would be faster to refuel FCEVs than BEVs in remote locations, provided that hydrogen fuel storage can be made available on or near those sites. An off-road FCEV configuration is presented by Saeks et al. [32], where a flywheel energy storage system [1] was used to recover energy and to aid in acceleration. The system had four motors in each of the four wheels to provide four-wheel drive, and employed adaptive controllers with interconnections to facilitate front- and rear-wheel steering as well as energy management and acceleration–deceleration. The works reviewed in this subsection are summarized in Table 2.

**Table 2.** Academic literature overview of general off-road EV architecture.

| Reference | Year | EV Type | Components of Interest | Control Algorithm | Potential Vehicle Application | Implementation Level |
|---|---|---|---|---|---|---|
| Saeks et al. [32] | 2002 | FCEV | • Fuel cell<br>• Flywheel<br>• Electric motor | • Neural adaptive controller<br>• Adaptive dynamic programming controller | • Off-road driving | Simulation |
| Zhang et al. [27] | 2008 | Parallel PHEV | • Battery<br>• Electric motor | • Fuzzy logic | • Light off-road driving | Simulation |
| Baronti et al. [31] | 2013 | General | • LiFeO$_4$ battery management system | - | • Construction<br>• Agriculture | Simulation |
| Parsons et al. [18] | 2014 | Series HEV | • Diesel generator<br>• Hub-mounted electric motor<br>• 2-speed transmission<br>• Battery | - | • Military<br>• Construction | Simulation and Hardware implementation |
| Sinkko et al. [30] | 2014 | HEV | • Permanent magnet synchronous motor<br>• 2-speed transmission | - | • Construction<br>• Agriculture | Simulation |

### 2.2. Construction Equipment

This subsection is focused on electrification efforts on construction equipment. Special attention is paid to construction equipment with higher population or carbon dioxide (CO$_2$) emission contribution in California, USA, according to data from the California Air Resources Board (CARB) [24]. The lists of construction equipment types in 2018 as sorted by population and CO$_2$ emission are shown in Figure 1. These two lists are not necessarily the same, as some types of equipment tend to have larger engine sizes, which produce more CO$_2$ emission per hour. Moreover, some equipment types are used more than others. It is notable that off-highway trucks had a small population (ranked 12th) but were the third largest contributors of CO$_2$ emission among all the construction equipment types. Thus, efforts to electrify this type of construction equipment could yield significant CO$_2$ emission reduction. In this subsection, the review will concentrate on the top equipment types in terms of CO$_2$ emission contribution, namely, loader, tractor–loader–backhoe, excavator, off-highway truck, and scraper.

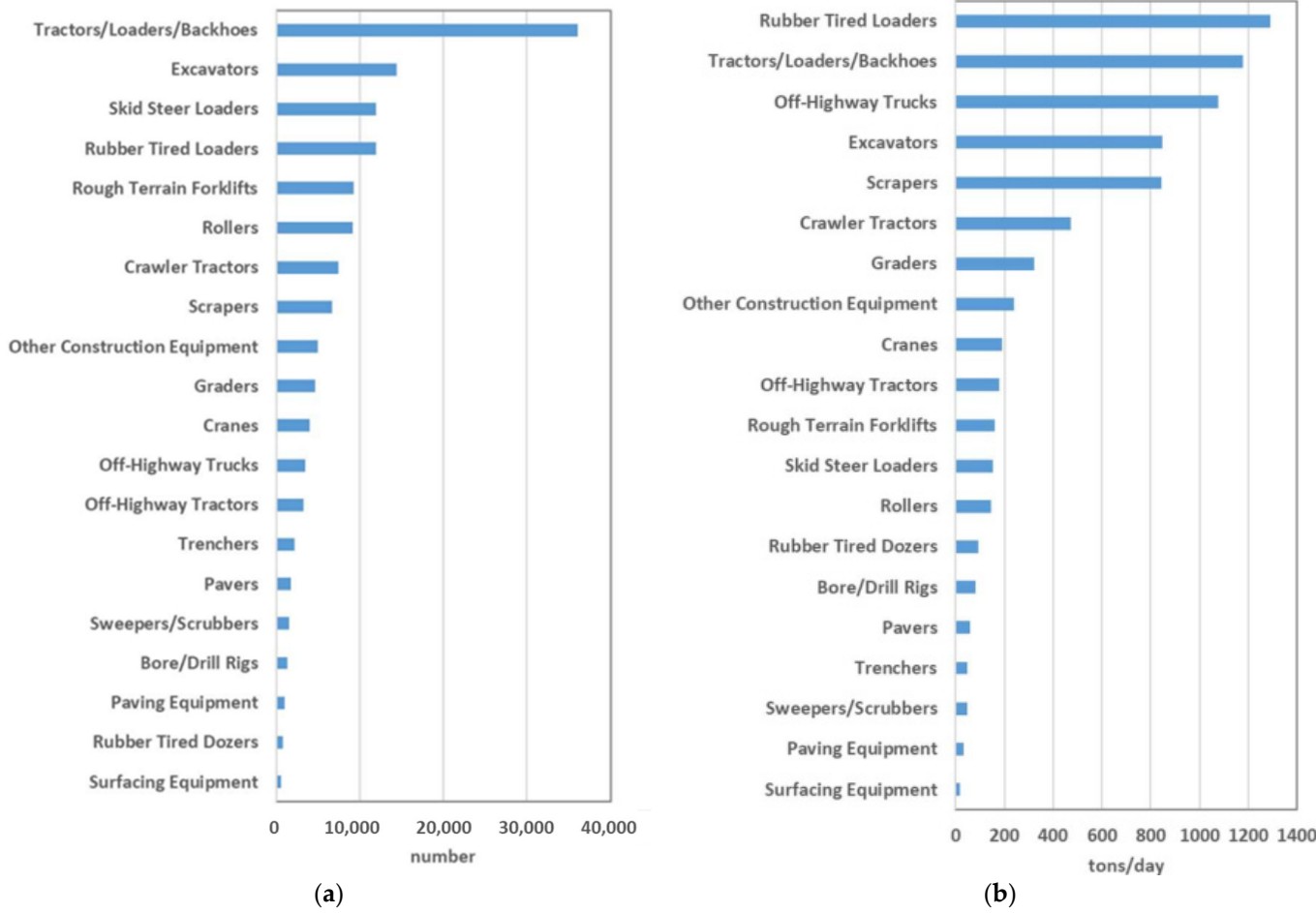

**Figure 1.** (**a**) Population and (**b**) CO$_2$ emission of various construction equipment types in California in 2018 (adapted from [24]).

Tractor–loader–backhoe (also known as backhoe–loader) is a tractor with a loader at the front and a backhoe at the back (Figure 2). Escorts [33] proposed a concept of an electric backhoe–loader, but details are currently limited [34]. Skid steer loaders are generally small, and can be tracked or wheeled. On the other hand, rubber-tired loaders are typically larger, and have articulate frames to allow the front wheels to pivot relative to the rear. Hybrid rubber-tired loaders are already available commercially [35–38], while BEV versions of skid steer loaders have also been introduced [38]. An example of hybrid rubber-tired loaders is the Caterpillar® 988K XE [39,40], which combines a switched reluctance electric drive with a Tier 4 diesel engine [41] for increased efficiency and convenience. It utilizes the switched reluctance machines as a generator and pump drive. Additional hybrid loader designs were reported in Achten et al. [42]. In addition, there has been development of BEV loaders, such as the Caterpillar R1300G LHD [43], which uses electric motors and lithium ion batteries to run the mechanical drivetrain with gears. Caterpillar also developed a commercial product, the R1700 XE LHD, which is shown in Figure 3.

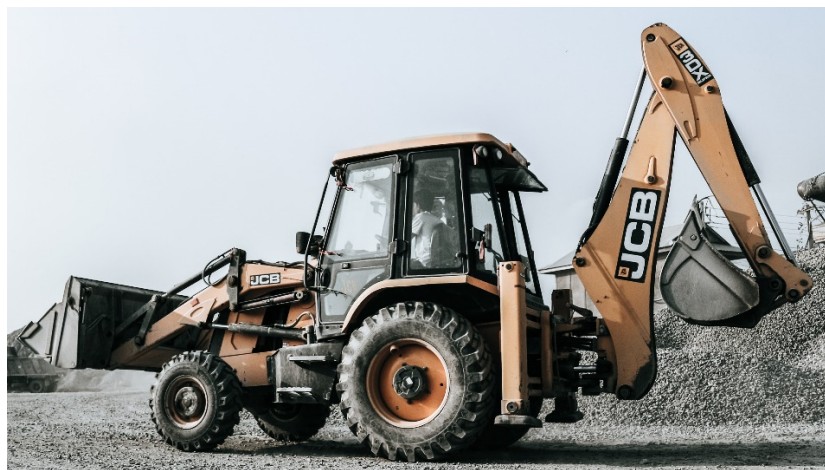

**Figure 2.** Tractor–loader–backhoes are tractors with a front-mounted loader and a back-mounted backhoe as attachments.

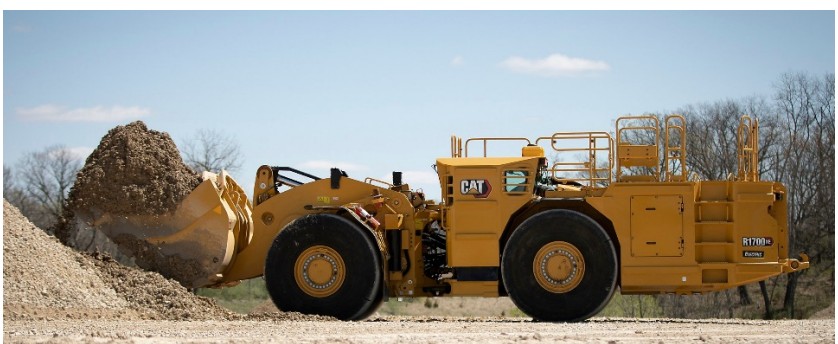

**Figure 3.** Caterpillar R1700 XE LHD which uses battery-powered electric motors for propulsion [44].

Excavators are fitted with digging equipment using a boom, and can be wheeled or tracked. Figure 4 shows an excavator. In [45,46], electric systems were successfully integrated into excavator booms for energy recovery, resulting in less energy consumption, and hence lower $CO_2$ emissions. Wang et al. [16] studied different drivetrain configurations for hybrid excavators. They found that a combination of electric motor with battery was most frequently used for small hybrid excavators, whereas medium hybrid excavators favored supercapacitors (SC) (also known as ultracapacitors) instead of battery as the ESS. The superior power density of SC and its faster power transfer in larger amounts as compared to battery might have driven this choice. The use of battery in hybrid excavators was also documented in Xiao et al. [47], while the use of hybrid ESS comprised both battery and SC was also proposed [16,48,49]. Yao and Wand [50] proposed a hybrid excavator using a supercapacitor to power its electric swing system. Kwon et al. [20] classified hybrid excavators in three configurations: series (electric motor controls all movements, powered by ICE), parallel (both ICE and motor powers the system), and compound (electric motor replaces the hydraulic swing motor facilitating energy recovery). They determined the compound system to be superior because of its greater reliability and shorter anticipated payback period. They also proposed a power control algorithm for compound hybrid excavators, which was claimed to reduce fuel consumption by 24% as compared to conventional excavators. This algorithm works by balancing power demand between the supercapacitor and the engine at each instance. In this hybrid configuration, the supercapacitor, the swing motor, and the generator (powered by the engine) are all connected to a pulse width modulation (PWM) converter (Figure 5). The power balance is attained by controlling this converter's DC-link voltage. The generator maintains a constant DC-link voltage utilizing a feedback mechanism, and the supercapacitor voltage

is kept in a certain range through a feed-forward mechanism while the engine speed is kept almost constant. The hydraulic pump is driven by the generator, which is run by the engine. According to some operational set points, the system power is supplied or absorbed (during swing regeneration) by either the generator or the supercapacitor. When the supercapacitor voltage is within its rated operational range, it is used to power the swing, and the generator charges the supercapacitor. In such a scenario, the supercapacitor also absorbs any regeneration from the swing. If the supercapacitor voltage is higher than the rated value (indicating that it cannot absorb any more energy), regeneration from the swing is used to run the generator in motoring mode, thus sharing the hydraulic load with the engine. In cases of zero swing power with a high supercapacitor voltage, the supercapacitor is discharged to share the hydraulic load with the engine by running the generator in motoring mode.

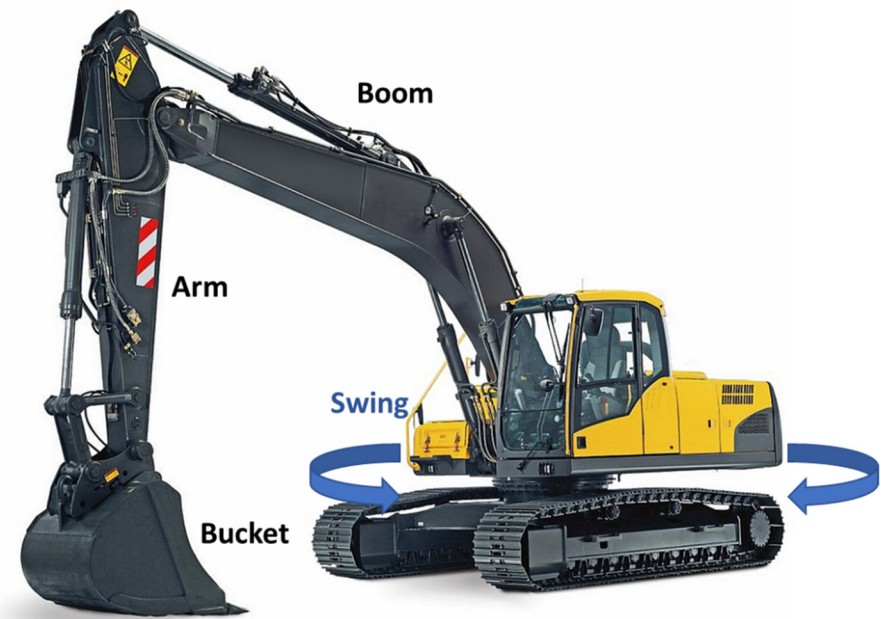

**Figure 4.** A wheeled excavator shown with its major components. The swing motion allows this equipment to rotate 360 degrees without engaging the drivetrain.

Wang et al. [21] also conducted a comparative study of hybrid excavator configurations, and identified the parallel system to be the best based on cost and performance considerations. Although they did not explicitly consider a compound system, the compound hybrid configuration in Kwon et al. [20] can be considered as a part of the parallel configuration set in Wang et al. [21], thus supporting the argument about the superiority of this configuration. A similar conclusion was made by Lin et al. [51] as well. Lee et al. [52] simulated a plug-in hybrid excavator in series, parallel, and compound modes, and the model showed that the compound mode could exploit the benefits of both series and parallel configurations but with higher cost and complexity. Yoo et al. [53] developed a hybrid control system with SC to operate in series, parallel, and compound modes, and then implemented the control system in a mid-sized excavator successfully. Xiao et al. [54] presented a control strategy for a parallel hybrid excavator employing ICE and SC to dynamically control the ICE's operating region for better overall system operation with little effect on performance. Ge et al. [55] used a variable speed electric motor to drive a variable displacement pump to meet the dynamic energy demand of excavators, which resulted in 1.35 kW less power consumption during idling and around 30% energy savings as compared to a pure displacement variable design.

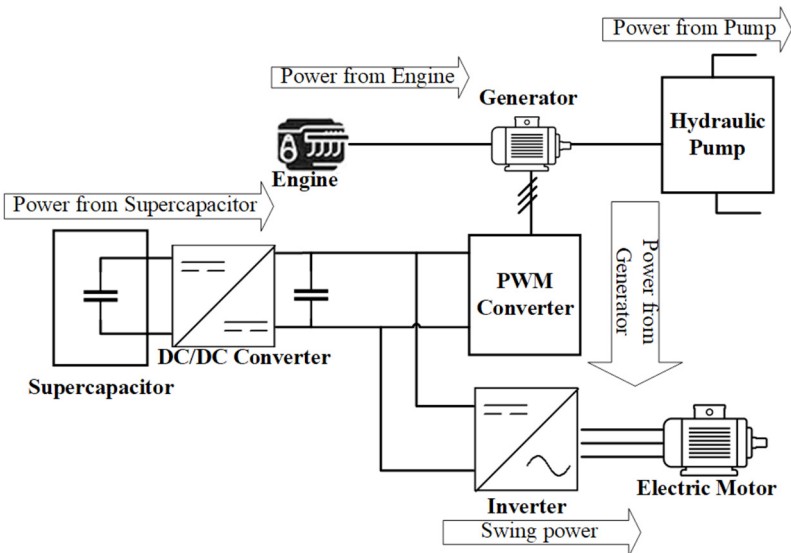

**Figure 5.** Configuration of compound hybrid excavator [20]. The supercapacitor is used as the electrical energy storage system while the electric drivetrain runs the swing electric motor with engine assistance.

Off-highway trucks (Figure 6) are also known as mining haul trucks [56]. Many them use diesel-electric drivetrains (electric drivetrains without high-voltage storage, powered by diesel engines [23]) with dynamic braking that employs AC wheel motors [56–58]. Efforts have been made to recover the braking energy, which is generally sent to brake resistors to be dissipated as heat (hence, the term dynamic braking) by adding ESS. This essentially transforms the diesel-electric architecture into a series hybrid one. Such an attempt was made by Richter et al. [56], where they successfully implemented a Sodium-Nickle-Chloride (NaNiCl$_2$) battery ESS in a Komatsu 830E [55]. Mazumdar [59] presented a truck trolley system where the trucks were provided with electricity from a dedicated substation through an overhead line to make the vehicles all-electric, thus reducing the fuel consumption even more by transferring the ICE's power generation operation to a more efficient system (the electrical grid). In this work, the use of supercapacitors was also proposed to capture regenerated energy for use in stretches of track where overhead lines could not be placed. Esfahanian et al. [57] proposed the use of road-grade data to dynamically control the energy management system (EMS) of a hybrid mining haul truck with ESS. This approach allowed the battery level to drop below safe state of charge (SoC) limits if there were downhill slopes within reach, which could replenish the battery and bring the SoC level back within the safe operating window through regenerative braking. The use of an AC–AC converter to run the AC motors in off-highway trucks without an intermediate DC converter was proposed by Kwak et al. [60], where they presented a matrix converter architecture with phase redundancy that came with fault detection capabilities. There have also been pilot projects demonstrating battery electric mining haul trucks. An example is a Komatsu 605-7 truck retrofitted with a 700 kWh Lithium Nickel Manganese Cobalt Oxide (LiNiMnCo) (called NMC in industry-standard nomenclature) battery pack and a synchronous motor [61]. Additionally, Mirzaei et al. [62] presented software and hardware solutions for improved electric braking in such trucks, where the hardware solution was proved to be more reliable but more costly than the software one. From the review of literature, it is evident that the diesel-electric powertrain has been widely used in off-highway trucks. Recent research in this area has focused on technologies to further electrify these trucks, such as integrating ESS for capturing energy from regenerative braking, and employing overhead power lines for full-electric operation.

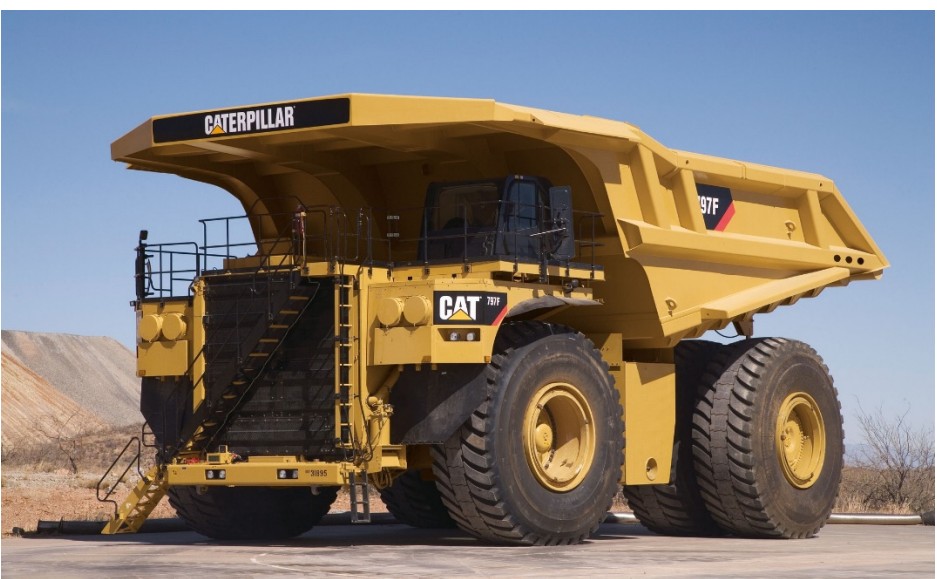

**Figure 6.** Off-highway truck.

According to the literature reviewed in this subsection, a significant amount of effort has already been made in hybridizing excavators, as evidenced by a large body of research work reviewed in Section 2.2. The reason behind there being this much interest in hybrid excavators is comprehensible. Hydraulic excavators are one of the most used pieces of construction equipment [63]. Their energy consumption is vast, yet the efficiency of converting that energy to useful work is quite low—less than 30% if fuel-to-actuator efficiency is calculated. Pollutant emissions including particulate matter (PM) and nitrogen oxides ($NO_x$) from this type of equipment is very high as well. The primary reason behind these is that the ICE is often operated near its rated speed, as opposed to in the high-efficiency region, so that the hydraulic pressure stays at a sufficient level to facilitate smooth transition from light to heavy load [20]. Moreover, the hydraulic system itself has an average efficiency of around 54% [55]. Thus, hybridization can significantly improve fuel efficiency and reduce the emission of hydraulic excavators, as the electric motor can help supply the instantaneous power required, letting the ICE operate in its most efficient region. In addition, the electric motor coupled with ESS can capture and store regenerative power, which is wasted as heat in excavators with ICE [20,64]. A similar observation can be made for off-highway trucks where the diesel-electric system has become mainstream, and series hybrid as well as battery electric options are being considered. On the other hand, the electrification of other major types of construction equipment, such as tractor–loader– backhoes, rubber-tired loaders, and scrapers, has not received the same level of attention. As they are major emitters of $CO_2$, PM, and $NO_x$, increased research and development effort to electrify these types of construction equipment is warranted. The academic and industrial works reviewed in Section 2.2 are summarized in Tables 3 and 4, respectively.

**Table 3.** Academic literature on electric off-road construction equipment.

| Reference | Year | EV Type | Components of Interest | Control Algorithm | Implementation Level | Equipment Type |
|---|---|---|---|---|---|---|
| Kwon et al. [20] | 2010 | HEV | • ICE<br>• Electric generator<br>• Electric motor<br>• Supercapacitor<br>• Hydraulic pump | Balancing power demand between a supercapacitor and the engine at each instance. | Simulation | Excavator |
| Yao et al. [50] | 2013 | HEV | • ICE<br>• Permanent magnet synchronous motor<br>• Supercapacitor<br>• Electric swing system | Combination of proportional (P) controller and mixed sensitivity controller. | Simulation and Hardware implementation | Excavator |
| Xiao et al. [54] | 2008 | Parallel HEV | • ICE<br>• Electric motor<br>• Supercapacitor<br>• Hydraulic pump | Dynamic work point. | Simulation | Excavator |
| Lin et al. [51] | 2008 | Parallel HEV, Series HEV | • ICE<br>• Electric motor<br>• Hydraulic pump | Dynamic multi work point controller comprising of direct torque control, and closed loop proportional-integral (PI) control. | Simulation | Excavator |
| Lee et al. [52] | 2013 | Parallel, series, and dual mode power split PHEV | • ICE<br>• Electric generator<br>• Electric motor<br>• Battery<br>• Hydraulic pump | Electric motor drives hydraulic pump, powered by battery; battery is charged by the generator run by ICE. | Simulation | Excavator |
| Yoo et al. [53] | 2009 | Parallel, series, and compound HEV | • Diesel ICE<br>• Electric motor<br>• Electric generator<br>• Electric swing motor<br>• Supercapacitor | Electric swing system, electric power assistance of ICE, regenerated energy stored in SC. | Simulation and hardware implementation | Excavator |
| Ge et al. [55] | 2017 | HEV | • ICE<br>• Speed variable electric motor<br>• Variable pump | Variable speed electric motor drives a variable displacement pump to meet the dynamic energy demand. | Simulation and hardware implementation | Excavator |

Table 3. *Cont.*

| Reference | Year | EV Type | Components of Interest | Control Algorithm | Implementation Level | Equipment Type |
|---|---|---|---|---|---|---|
| Wang et al. [65] | 2013 | HEV | • ICE<br>• Electric generator<br>• Electric motor<br>• Supercapacitor<br>• Potential energy recovery system<br>• Electric swing system | Energy regeneration from swing system and boom. | Simulation | Excavator |
| Mazumdar [59] | 2013 | BEV | • Electric drivetrain<br>• Overhead power line<br>• Regenerative braking<br>• Battery or SC energy storage system (ESS) | Driven by overhead power supply. Regenerated energy stored in ESS to use in short driving distances. | Simulation | Off-highway truck |
| Esfahanian et al. [57] | 2013 | HEV | • ICE<br>• Electric motor<br>• Battery<br>• Regenerative braking | Road grade data used for dynamic energy management. | - | Off-highway truck |

Table 4. Industrial research on electric off-road construction equipment.

| Reference | Manufacturer | Model | EV Type | Components of Interest | Control Strategy | Equipment Type | Implementation Level |
|---|---|---|---|---|---|---|---|
| [36,66] | John Deere | 644K Hybrid Wheel Loader | HEV | • Interim tier 4 diesel engine<br>• 3-phase alternating current (AC) motor/generator<br>• Water-cooled inverter<br>• Water-cooled brake resistor<br>• Battery | No reverse gear as electric motor can perform this shift in direction, brake resistor consumes and dissipates excess energy generated during regenerative braking. | Skid steer loader/rubber-tired loader | Hardware implementation |
| [37,38] | John Deere | 318E<br>320E<br>326E<br>328E<br>332E | HEV | • Final/Interim tier 4 diesel engine<br>• Electrohydraulic powertrain | - | Skid steer loader/rubber-tired loader | Hardware implementation |
| [39] | Tobroco-Giant | GIANT E-skid steer | BEV | • Hydraulic wheel motor<br>• Battery | - | Skid steer loader/rubber-tired loader | Hardware implementation |

**Table 4.** *Cont.*

| Reference | Manufacturer | Model | EV Type | Components of Interest | Control Strategy | Equipment Type | Implementation Level |
|---|---|---|---|---|---|---|---|
| [44] | Caterpillar | R1300G LHD | BEV | • Lithium battery pack<br>• Electric motor<br>• Mechanical axles and drive-shafts | Electric motor used to run mechanical drivetrain through electric motor. | Rubber-tired loader | Hardware implementation |
| [40,41] | Caterpillar | 988K XE | HEV | • Tier 4 diesel engine<br>• Switched reluctance electric machine for drivetrain, pump drive, and generator<br>• Specialized power electronics | - | Rubber-tired loader | Hardware implementation |
| [16] | Kobelco (modified) | 70SR | HEV | • 288 Volt Li-ion battery set<br>• 20 kW electric motor/generator<br>• Electric swing | Energy supplied to the electrical load from the battery when needed, and absorbed during braking. | Excavator | - |
| [16,67] | Kobelco | SK80H | HEV | • 288 Volt nickel metal hydride battery set<br>• 20 kW electric motor/generator<br>• 10 kW electric swing motor | Battery charging and discharging limit set according to concurrent state-of-charge to ensure maximum efficiency and lifetime. | Excavator | Simulation |
| [16] | Caterpillar | - | Parallel HEV | • ICE<br>• Electric motor/generator<br>• Battery | Operating mode and torque set according to load variation and SOC. | Excavator | - |
| [16] | Komatsu | - | HEV | • ICE<br>• Electric generator<br>• Electric motor<br>• Supercapacitor<br>• Electric swing system | Separate use of hydraulic motor and generator. | Excavator | - |

**Table 4.** *Cont.*

| Reference | Manufacturer | Model | EV Type | Components of Interest | Control Strategy | Equipment Type | Implementation Level |
|---|---|---|---|---|---|---|---|
| [16,67] | Hitachi | - | Parallel HEV | • ICE<br>• Electric generator<br>• Electric motor<br>• Supercapacitor<br>• Electric swing system | Control system comprised of master and slave controllers where the slave is used to monitor and govern the SC charge-discharge. | Excavator | - |
| [16] | Doosan | - | HEV | • ICE<br>• Electric generator<br>• Electric motor<br>• Supercapacitor | - | Excavator | - |
| [16,48,49] | Kobelco | - | Series HEV | • ICE<br>• Hybrid ESS (288 V, 6.5 Ah Ni-MH battery + 304 V, 11.4 F SC) | ESS assists during heavy load and stores surplus energy under light loads.<br>Engine works in high efficiency region all the time, even stops when ESS energy is sufficient to drive loads. | Excavator | - |
| [16] | Sumitomo | - | HEV | • ICE<br>• Supercapacitor<br>• Electric motor | SC SoC set to a higher value to drive load at higher voltage with better efficiency. | Excavator | - |
| [56] | Komatsu | 830E (modified) | Series HEV | • ICE<br>• $NaNiCl_2$ battery<br>• Wheel motor | Battery used to recover braking energy to be deployed for power boost or enhanced engine efficiency. | Off-highway truck | Simulation and hardware implementation |
| [58] | Komatsu | 830E-1AC | Series HEV | • Tier 2 Diesel engine<br>• Electric generator<br>• Wheel motor<br>• Electric retarder (dynamic) | - | Off-highway truck | Commercially available |
| [68] | Komatsu | 930E-4 | Diesel-electric with dynamic braking | • Tier 2 Diesel engine<br>• Electric generator<br>• Wheel motor<br>• Electric retarder (dynamic) | - | Off-highway truck | Commercially available |

**Table 4.** *Cont.*

| Reference | Manufacturer | Model | EV Type | Components of Interest | Control Strategy | Equipment Type | Implementation Level |
|---|---|---|---|---|---|---|---|
| [69] | Caterpillar | 795F AC Mining Truck | Diesel-electric with dynamic braking | • ICE<br>• Electric generator<br>• AC induction wheel motor<br>• Electric retarder (dynamic) | - | Off-highway truck | Commercially available |
| [60] | Komatsu | 605-7 (modified) | BEV | • LiNiMnCo battery pack<br>• Synchronous motor<br>• Regenerative braking | The battery powers the motor and stores regenerative energy. | Off-highway truck | Hardware implementation |

### 2.3. Agricultural Equipment

This subsection focuses on electrification attempts on agricultural equipment. Similar to construction equipment, special attention is paid to equipment with higher population or carbon dioxide ($CO_2$) emission contribution in California, USA, according to data available from the California Air Resources Board (CARB) [24]. Agricultural equipment types recorded in the CARB database are shown in Figure 7, sorted by population and $CO_2$ emission in 2018. As mentioned in the previous subsection, these two lists showing population and emission are not necessarily the same, because of larger engine sizes and/or higher use of some equipment types—which led to greater $CO_2$ emission per hour despite their lower population. It can be seen from Figure 7 that agricultural tractors have much smaller population than tillers, but supersede them in terms of $CO_2$ emission. Tractors are also identified as the most fuel-consuming mobile agricultural equipment [70], which provides some explanation of their higher $CO_2$ emission. Thus, electrifying the agricultural tractors could yield significant $CO_2$ emission reduction, and this subsection will concentrate on this single agricultural equipment type. Figure 8 shows an agricultural tractor.

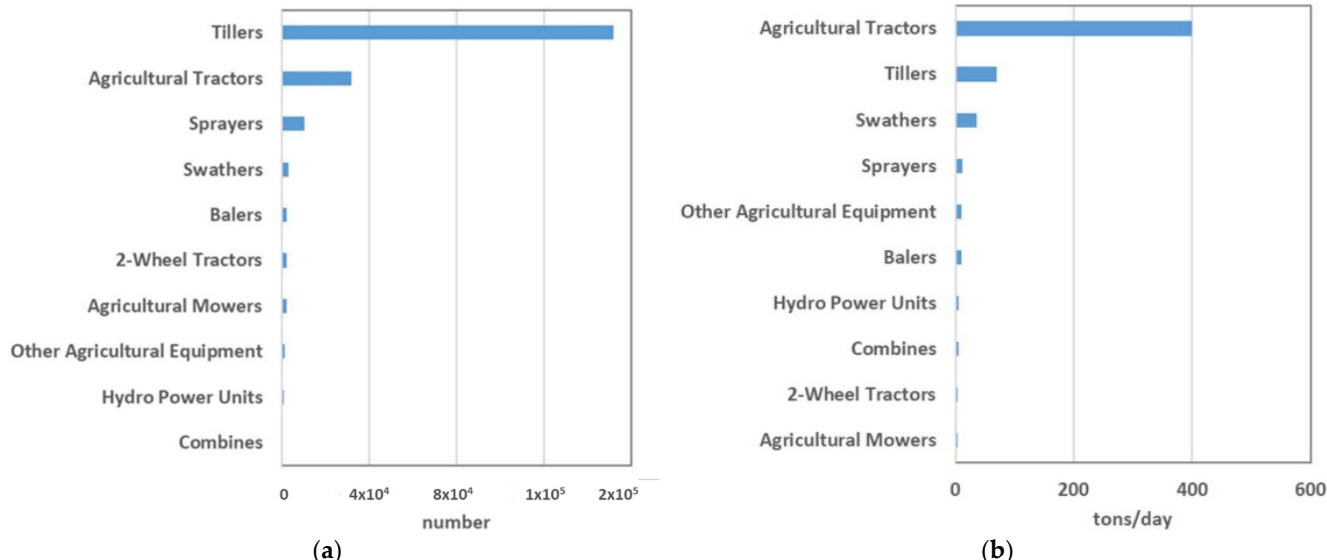

(**a**)                      (**b**)

**Figure 7.** (**a**) Population and (**b**) $CO_2$ emission of various agricultural equipment types in California in 2018 (adapted from [24]).

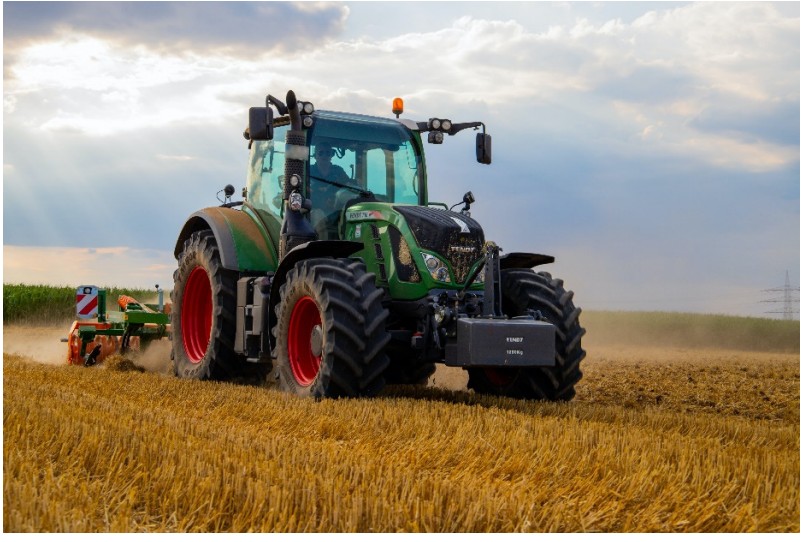

**Figure 8.** Agricultural tractor. Series hybrid concept for this type of equipment was presented in [71] with electric drivetrain and PTO.

Previously, Usinin et al. presented a series hybrid electric drivetrain for tractors, having an engine, generator, two traction motors, and required power electronics [72]. Gas turbine and diesel engines were proposed as the engine choices; while electric machines and power electronics were designed to reduce cost [72]. Mousazadeh et al.'s design employed two solar panels on their tractor, which was capable of meeting 18% of the energy demand, and the rest was obtained from the grid to charge its valve-regulated lead acid (VRLA) battery pack [73]. This tractor successfully carried out several common light agricultural tasks, including plowing, mowing, and towing. This equipment was mentioned as a PHEV, but based on the definitions used in this paper, it was a BEV because of its sole use of electric drivetrain and absence of ICE. It is categorized accordingly in Table 5. Mousazadeh et al. conducted a comparative study on different battery technologies best suited for their solar-assisted tractor in [74]. They concluded that the VRLA technology was the best considering the regional manufacturing capabilities. Ueka et al.'s design used an electric motor to drive a rotary tiller and employed four wheel drive in a battery electric tractor [70]. An electronically controlled continuously variable transmission (e-CVT) with PTO capabilities was designed and implemented by Rossi et al. for a parallel hybrid agricultural tractor [75]. Florentsev et al. presented a pre-production version of a series hybrid tractor. It used an asynchronous traction motor and electricity-driven PTO [71]. A similar work was shown by Puhovoy et al. in [76]. To enable high-voltage PTO capabilities, Moreda et al. proposed installing a PTO-dedicated high voltage generator on tractors [23]. Gonzalez-de-Soto et al. presented a hydrogen-fuel-cell-powered PTO system for an ICE-driven tractor [25]. Their system comprised a fuel cell stack and a solar photovoltaic (PV) system for power generation, and batteries for storage. A fuel cell electric tractor was also demonstrated previously [77]. Additionally, Zhitkova et al. designed an electric motor for agricultural tractor use. This motor was suited for both low speed off-road operation and higher speed produce-transportation work [78].

The academic and industrial works reviewed in Section 2.3 are summarized in Tables 5 and 6, respectively. Figure 9 shows an infographic of the works reviewed in different subsections of Section 2.

**Table 5.** Academic literature on electric off-road agricultural equipment.

| Reference | Year | EV Type | Components of Interest | Control Algorithm | Implementation Level | Equipment Type |
|---|---|---|---|---|---|---|
| Usinin et al. [72] | 2013 | Series HEV | • Gas turbine/diesel ICE<br>• Synchronous reluctance generator<br>• Synchronous reluctance motor | Separate excitation for generator and motor, motor torque control by controlling armature current and magnetic flux. | Simulation | Tractor |
| Mousazadeh et al. [73] | 2010 | BEV | • VRLA battery pack<br>• Electric motor<br>• Solar panel<br>• Electrically driven PTO | Solar panel supplied 18% of required power, rest taken from grid. | Simulation and hardware implementation | Tractor |
| Ueka et al. [70] | 2013 | BEV | • Battery pack<br>• Electric motor<br>• Electrically driven PTO | A rotary tiller along with the four wheels driven by the motor through reduction gear. | Simulation and hardware implementation | Tractor |
| Rossi et al. [75] | 2014 | Parallel HEV | • ICE<br>• Electric motor/generator<br>• e-CVT with PTO | Set up for using ICE's maximum torque operating region. | Simulation and hardware implementation | Tractor |
| Gonzalez-de-Soto et al. [25] | 2016 | ICE vehicle with fuel cell-powered PTO | • ICE<br>• Hydrogen fuel cell<br>• Solar photovoltaic system<br>• Battery | The fuel cell system powers the PTO, while ICE runs the drivetrain. Battery stores excess energy. | Simulation and hardware implementation | Tractor |

**Table 6.** Industrial research on electric off-road agricultural equipment.

| Reference | Manufacturer | Model | EV Type | Components of Interest | Control Strategy | Equipment Type | Implementation Level |
|---|---|---|---|---|---|---|---|
| [71] | Ruselprom | Belarus-3023 | Series HEV | • ICE<br>• Battery<br>• Liquid-cooled asynchronous motor/generator<br>• Liquid-cooled asynchronous traction motor<br>• Liquid-cooled power electronics<br>• Electric-powered PTO | ICE powered electric drivetrain, electricity driven PTO. | Tractor | Pre-production versions produced |
| [77] | New Holland | NH2 | FCEV | • Fuel cell stack<br>• Electric motors for traction and PTO | Traction and PTO operation handled by separate motors. | Tractor | Hardware implementation |

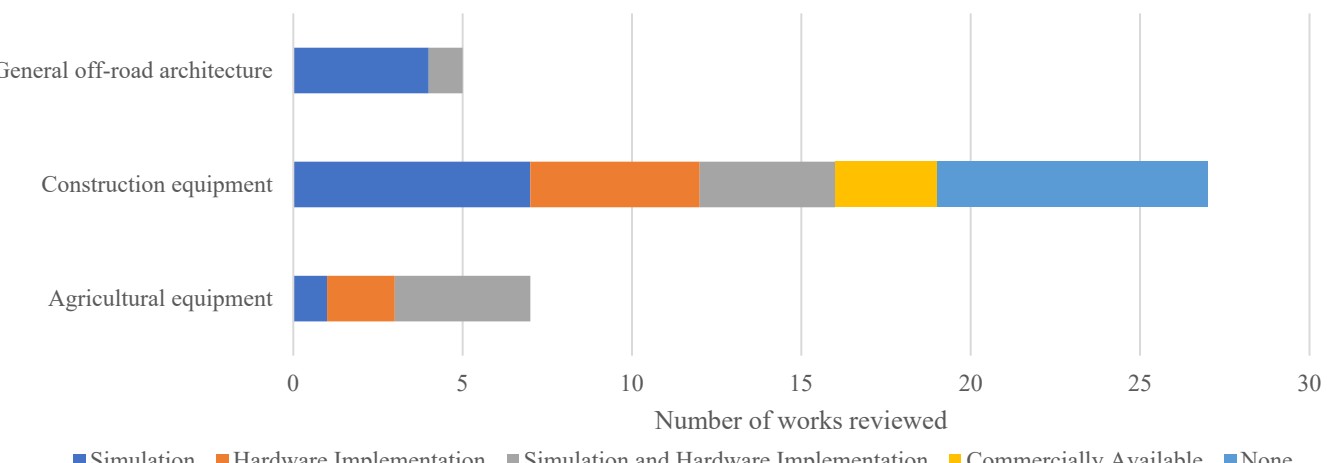

**Figure 9.** Comparative visualization of major works reviewed in Section 2: in the works conducted so far, emphasis has been given on both simulation and hardware implementation—both of them in many cases. For the construction sector, several commercially available vehicles fell into the interest of this review.

### 3. Energy Recovery

In addition to the commonly used regenerative braking employed for on-road electric vehicles [1], off-road equipment can utilize other methods for energy recovery, such as regeneration in excavators from the swing and boom movement [43]. This section describes the regeneration methods observed for the studied equipment types.

Loaders stop abruptly during operation for piling material, then lifting, moving, and dropping those. These stops can generate electricity through regenerative braking [79,80]. This strategy was implemented in the John Deere 644K Hybrid Wheel Loader [36,66]. Kinetic energy recovery in loaders through regenerative braking is less compared to on-road vehicles as loaders operate with greater rolling resistance. Electric retarders in off-highway trucks conduct braking by dissipating energy as heat; capturing this energy in ESS was proposed in [58,59]. Potential energy can be used to generate electricity to be stored in ESS while lowering forklift-type systems [81].

Potential energy can be captured while lowering the boom of an excavator. Yoon et al. proposed an ESS consisting of battery and capacitors to capture this energy [45]. Ge et al.'s method could capture energy as hydraulic energy [46]. Another hydraulic energy capture system was presented by Ho et al. in [82]. Xia et al. also presented a hydraulic potential energy recovery method applicable to machines with hydraulic cylinders [83]. Though such methods do not generate electric energy directly, these can still be useful in hybrid equipment where hydraulic systems work alongside the electrical powertrain. Lin et al. noted that with an electric recuperation system directly coupled with an excavator boom, the regeneration time-window got directly related to the duration of lowering the boom, which could be too short for a battery to capture all the available energy; moreover, the electric generator had to work at different efficiency points if the load point shifted, lowering overall efficiency. To counter this, they proposed using a hydraulic cylinder for the fast capture of the potential energy, and then used it to run a generator to efficiently store the electricity energy in an ESS. They used supercapacitors for this purpose, but mentioned that the use of batteries was also possible, as the intermediate hydraulic cylinder can facilitate the fast-capture of energy and then run the generator for a period best-suited for the battery to charge properly [84]. These justifications were supported in [85], where a hydraulic motor/generator was used to capture energy from a parallel hybrid excavator's boom and store in an ESS. It also pointed out that without the intermediate hydraulic system, even using supercapacitors as ESS would be unwise, as the instantaneous large changes in power could affect the lifetime of the supercapacitors used. It was also

identified there that the boom was the major source for regenerative energy in the 7-ton excavator used in that work, as 67% of total recapturable energy came from its movements. Wang et al.'s method also proposed to couple an electric generator with hydraulic cylinders for electricity generation from cylinder pressure, which could be consumed instantly by some other operating components, or stored in ESS for future use [86]. Chen et al. showed a method for capturing gravitational energy from excavator booms by running a permanent magnet brushless direct current (DC) motor, and storing the energy in supercapacitors [87]. Yoo et al. proposed energy regeneration and subsequent storage in supercapacitor from the swing movement in [53], whereas Wang et al. opted for recuperation from both swing and boom [65].

Other than these, generation of electricity by recapturing heat from turbocharged engines could be achieved by running the exhaust gas leaving the turbocharger through a second turbine-generator system, or by using thermoelectric generators—which do not require any moving parts for the generation [23,79,88]. Therefore, such techniques can potentially be applied to any construction or agricultural equipment employing a turbocharger. Additionally, electro-hybrid actuators for off-highway equipment were proposed by Åman et al. to replace hydraulic pipelines with electric wiring, thus enhancing reliability, and also facilitating regeneration from hydraulic systems [89].

The key technologies reviewed in this section are listed in Table 7. From this section, it is evident that most of the studies conducted were on excavators—consistent with the findings in Section 2. Other equipment types (tractor–loader–backhoe, loader, off-highway truck, scraper, and agricultural tractor) received limited or no attention.

**Table 7.** Energy recovery techniques in reviewed literature for equipment types of interest.

| Reference | Year | Regenerative Component | Vehicle Application | Implementation Level | Equipment Type |
|---|---|---|---|---|---|
| Minav et al. [81] | 2013 | Lift | Construction | Simulation | Forklift |
| Mazumdar [58] | 2013 | Brake | Construction | Simulation | Off-highway truck |
| Esfahanian et al. [57] | 2013 | Brake | Construction | - | |
| [60] | 2017 | Brake | Construction | Hardware implementation | |
| Yoon et al. [45] | 2013 | Boom | Construction | Simulation | Excavator |
| Wang et al. [86] | 2014 | Hydraulic cylinder | Construction | Simulation | |
| Lin et al. [84] | 2016 | Boom | Construction | Simulation and hardware implementation | |
| Lin et al. [85] | 2010 | Boom | Construction | Simulation | |
| Chen et al. [87] | 2017 | Boom | Construction | Simulation and hardware implementation | |
| Yoo et al. [53] | 2009 | Swing | Construction | Simulation and hardware implementation | |
| Wang et al. [65] | 2013 | Swing and boom | Construction | Simulation | |
| Singh et al. [80] | 2009 | Turbocharger | Construction Agriculture | - | All turbocharged equipment |
| Yu et al. [88] | 2015 | Turbocharger | Construction Agriculture | - | |
| Åman et al. [89] | 2013 | Electro-hybrid actuator | Construction Agriculture | Simulation | Off-highway equipment |

## 4. Promises and Concerns of Off-Road Equipment Electrification

This section looks into the advantages and existing challenges of off-road construction and agricultural equipment electrification. Some benefits and issues are shared with the

on-road vehicle segment, but there exist some unique ones because of the equipment's unique activity demands.

### 4.1. Advantages

Compared to ICEs, electric motors are more capable to meet high torque demands [26,29]. This can be useful for off-road equipment applications. Electric drivetrains usually have fewer moving parts than traditional ICE systems [1]. Regenerative braking also reduces wear on mechanical brakes [61]. Because of this, electric drivetrains experience less wear and reduced maintenance costs. Combined with reduced fuel consumption, this results in lower operating costs. An electric drivetrain increases the powertrain efficiency overall, for both hybrid and full electric configurations. It also allows for the decoupling of loads from the ICE in some vehicles, such as agricultural tractors [23]. ICEs tend to lose power at high altitude because of insufficient oxygen required to burn fuel and generate power. Pieces of electric equipment do not suffer from this drawback. This can facilitate easier operation, better efficiency, and lower fuel cost in such operating conditions. The reduced downtime from lower maintenance requirement can result in higher productivity [80]. Electric vehicles also allow for more flexible design options [23], offering more space and better utilization of it. Vehicle electrification is facilitating unique operational and economic benefits as well. One such possibility is operating the equipment closer to emission-and-noise-sensitive areas and hours. This cannot be achieved with ICE equipment, and the use of electric ones can increase operating flexibility and productivity for such cases. Lower emissions can also be beneficial for underground operating scenarios, such as mines, where the air quality can be significantly improved if the equipment causes less air pollution [44]. These advantages, and their effects, are listed in Table 8.

**Table 8.** Advantages of equipment electrification and their implications.

| Advantage / Implication | Environmental | Operational | Economic |
|---|---|---|---|
| • Less moving parts<br>• Instant bidirectional torque<br>• Higher efficiency<br>• Electric deceleration<br>• No power loss at high altitudes | • Less emission | • Ease of operation<br>• Simpler drivetrain<br>• Less wear<br>• Less maintenance | • Less operating cost<br>• Less downtime<br>• Increased work efficiency and productivity |
| • Less fuel consumption | • Less emission<br>• Improved workplace environment | • Less dependency on fuel supply | • Less operating cost |
| • Reduced noise | • Reduced noise pollution | • More flexibility in choosing operating hours and areas | • Increased productivity<br>• Reduced downtime |
| • Flexible design | - | • More utility | • Potential reduction in manufacturing cost |

### 4.2. Limitations and Solutions

Major drawbacks of EVs include long charging time and short range [1]. These can cause shortened operating time and increased downtime for construction and agricultural equipment. Moreover, as the off-road equipment have far superior and dynamic power requirements, sizing of motor and ESS considering design constraints (e.g., weight) becomes a major design concern [16,26]. Facilitating charging for electric off-road equipment can also be challenging. For construction equipment, jobsites can be temporary and can move around. For agricultural equipment, wide operating areas can demand strategic placement of charging stations. However, recent technological developments improved range and reduced charging time. Moreover, though downtime for charging is a concern for operators,

this can be compensated by the reduction in maintenance downtime. Designing off-road EV powertrains to meet the dynamic power needs has already been tackled in several studies—the most common approach being the use of gears to satisfy the varying power demand with smaller motor sizes [18].

*4.3. Current Barriers*

The high price of EVs, and strong competition from conventional ICE-driven equipment can be considered as probable barriers for electrification in the off-road segment [80]. However, the trend towards partial electrification for some equipment types (e.g., excavators, off-highway trucks) can change that paradigm. Beyond these shortcomings inherent to the early stages of EV adoption, the lack of research for multiple equipment types can be considered as a major impediment for electrifying this sector. However, the commercial use of electric drives in off-highway trucks provides an example of electric powertrains' capability for off-highway applications. Now, increased research and development for electrification is required for other equipment categories (e.g., loaders, scrapers). Along with industry interest, government efforts in the form of regulations, incentives, and grants can play a major role in increasing electrification in these sectors. Such actions can compensate the higher cost of EVs. A nascent electric off-road equipment sector is likely to face difficulties with inadequate charging infrastructure as well. This can be addressed by manufacturers investing in developing charging infrastructure while marketing their products. Table 9 sums up the concerns surrounding off-road equipment electrification and their potential solutions.

**Table 9.** Concerns surrounding off-road equipment electrification and potential solutions.

| | Concern | Solution |
|---|---|---|
| Technical issues | Short range | • Better ESS<br>• Better energy recuperation techniques |
| | Long charging time | • High voltage charging |
| | Dynamic and high power requirement | • Use of transmission<br>• Improved ESS |
| Logistics issues | Lack of research | • Increased funding<br>• Regulations<br>• Incentives |
| | Inadequate charging infrastructure | • Development of necessary charging infrastructure while developing any commercial off-road equipment. |
| | Charging station placement | • Proper planning<br>• Mobile charging facilities |
| Market issues | Cost | • Increased production<br>• Lease<br>• Incentive |
| | Competition | • Regulations<br>• Incentives<br>• Proving superior performance |

**5. Proposals for Off-Road Equipment Electrification**

The pieces of off-road equipment studied have varied work environments and activity demands. In general, the use of these pieces of equipment depends a lot on respective duty cycles—which can vary for different jobsites. Because of this, it is not possible to make an individual optimal electrification recommendation that will be the most efficient for all off-road equipment. Moreover, as the jobsites vary in condition and duty cycles, equipment within the same category may benefit from different technologies depending on their intended use. This section thus lays out the general possibilities that can facilitate

off-road equipment electrification by overcoming the current limitations, but it is possible that effective application of these techniques can vary for each use-case.

Construction and agricultural equipment tend to have a significantly long service life, and fleet operators may not want to retire conventional equipment before its typical service lifespan. A plausible solution for such scenarios can be retrofitting the existing vehicles with an electric powertrain to utilize the remaining service life. One approach to retrofitting can be the use of range extenders [1] to act as an on-board generator. The use of the existing Tier 4 diesel engines as range extenders operating within optimal regions can maximize efficiency and minimize emissions while utilizing the existing lifecycle of these engines.

Operating PHEV or BEV equipment may require on-site charging facilities. One way to facilitate this can be to use renewable energy sources (RES) to power the chargers. Redpath et al. demonstrated the charging of light agricultural vehicles through solar energy [90], and similar scaled-up approaches can appear beneficial for heavy-duty agricultural equipment as well. The use of wind power for such cases can also appear useful [91]. Employing solar PV to charge EVs is a popular idea [92–94]. Bhatti et al. conducted a thorough study on this topic [95], where various such configurations were listed, including PV-fed EV charging stations with connection to the grid, with intermediate ESS, and dedicated fuel-cell generators. Robalino et al. proposed using PV to charge EVs while generating hydrogen at the charging station for FCEVs [96]. Such charging stations, equipped with ESS, and a hydrogen generating mechanism, can serve BEVs, PHEVs, and FCEVs, while utilizing all the generated electricity from the RES. Kam et al.'s proposed smart charging system with vehicle-to-grid (V2G) facility [97] can prove useful to realize energy-independent self-sustaining small agricultural farms. Second-life batteries (SLB) [98,99] can be employed in such charging stations as ESS to lower the cost. In the long term, this can become more efficient and cost-effective if SLBs from off-road equipment are used, extending the value of initial investments. Proper placement of charging stations, and the use of mobile chargers—which can power vehicles from energy stored in mobile ESS—can prove useful in cases where equipment cannot return to charging bases. This technology is currently available for passenger vehicles [100,101], with more expected to enter the consumer market soon [102]. Scaled-up versions of such devices can cater to heavy-duty equipment. Employing the FCEV architecture for off-road equipment can prove beneficial as well, as that will provide short refueling times similar to conventional vehicles—resulting in shortened downtimes. Some major reservations against fuel cells have been high price, and safety concerns regarding the on-board hydrogen tanks [1]. However, as the technology is getting more mature, and more commercial FCEVs are emerging [103,104], successful implementation of this technology in off-road equipment can be expected. Figure 10 presents the major proposals made in this section.

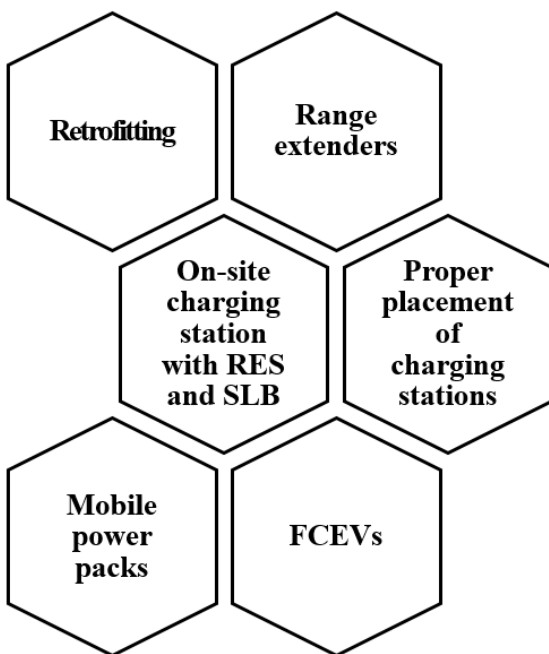

**Figure 10.** Potential technologies for facilitating off-road equipment electrification.

## 6. Outcomes and Future Works

The existing work in the off-road equipment sector addressed certain niches, and additional research is needed to facilitate electrification. The following points summarize the major findings of this paper to indicate the current state of this field, and the areas needing attention:

- Among the pieces of construction equipment, excavator and off-highway truck electrification attracted the most attention; efficiency gains and cost reduction have driven the commercialization of diesel-electric off-highway trucks.
- Tractors were studied in a number of reviewed studies on agricultural equipment.
- Tractor–loader–backhoes, loaders, and scrapers in the construction equipment category, and tractors from the agricultural equipment sector demand increased research on electrification potential due to their high population and impact on emissions.
- With current technology, hybrids can appear useful for immediate implementation.
- Along with batteries, supercapacitors attracted significant attention, as the equipment tends to have a high power requirement. For the same reason, intermediate hydraulic energy storage and hybrid energy storage employing batteries and supercapacitors can prove beneficial for heavy-duty equipment usage.
- Along with the braking system, there are opportunities for energy regeneration from power tools employed by off-road equipment; prominent examples being the boom and swing of excavators.
- Electrification of off-road equipment can offer significant benefits in terms of increased efficiency and lower operating cost.
- The general shortcomings of EVs, including short range and long charging time, can translate into concerns about decreased downtime for off-road equipment. The higher cost further challenges their acceptance in a competitive market. However, increased research and development can aid in overcoming the current issues.
- An immediate solution to facilitating successful electrification of off-road equipment is retrofitting along with the use of range extenders, on-site power generation, and mobile chargers.

Future work can be conducted on:

- Feasible existing and emerging technologies, and approaches for off-road equipment electrification considering the duty cycles, load factors, use case, and infrastructure requirement of different pieces of equipment.
- Ways to efficiently recapture energy in off-road equipment.
- Feasibility of mobile ICE and fuel cell generators for off-road equipment charging.
- Impact of regulations and incentives on the off-road equipment market.

## 7. Conclusions

The electrification of off-road construction and agricultural equipment is expected to improve operating efficiency while reducing operating cost and emissions. To provide a clear picture of the current state of these pieces of equipment, existing notable studies have been reviewed in this paper. The advantages and limitations for off-road equipment electrification have been discussed along with possible solutions. Proposals have been made to facilitate electrification attempts in this sector while underscoring the major findings and future research directions.

**Author Contributions:** Conceptualization, K.B., H.P., S.C. and S.Y.; Methodology, K.B. and F.U.-N.; Validation, K.B., G.W., H.P., S.C., S.Y. and M.B.; Formal Analysis, K.B. and F.U.-N.; Investigation, K.B. and F.U.-N.; Data Curation, K.B. and F.U.-N.; Writing—Original Draft Preparation, K.B. and F.U.-N.; Writing—Review and Editing, K.B., G.W., H.P., S.C. and S.Y.; Visualization, K.B. and F.U.-N.; Supervision,: K.B., G.W., H.P., S.C., S.Y. and M.B.; Project Administration,: K.B., G.W., H.P., S.C., S.Y. and M.B.; Funding Acquisition,: K.B., G.W., H.P., S.C. and S.Y. All authors have read and agreed to the published version of the manuscript.

**Funding:** This research was conducted in fulfillment of Agreement No. 18RD016 by the University of California, Riverside under the sponsorship of the California Air Resources Board (CARB).

**Acknowledgments:** The authors are thankful for the inputs from Christopher Weaver and Evan Powers of CARB. The statements and conclusions in this paper are those of the authors and not necessarily those of CARB. The mention of commercial products, their source, or their use in connection with material report herein is not to be construed of as actual or implied endorsement of such products.

**Conflicts of Interest:** The authors declare no conflict of interest.

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
