# Peer review of "Off-Road Construction and Agricultural Equipment Electrification: Review, Challenges, and Opportunities"

_vehicles, doi:10.3390/vehicles4030044_

Round 1

Reviewer 1 Report

This paper reviews the technical status of electrification of off-road equipment, focusing on the equipment used in construction and agricultural applications. This paper also discusses the advantages and challenges of electrification of off-road buildings and agricultural equipment. In addition, potential solutions to overcome these challenges and opportunities to promote off-road construction and electrification of agricultural equipment were identified.

I think the paper can be accepted and published.

Reviewer 2 Report

The paper is a well done review of the state of art of such vehicles.

I have some comments about layout and some suggestions about electrical motors used in these applications.

LAYOUT

Table I - new line after dot should be indented.

Table II - the four column changes in the pagination in wrong way.

Indent properly.

It is better to have Table II completely in the same page.

Fig1. caption of Fig.1 is in the page following the figure

Fig4. caption of Fig.4 is in the page following the figure

Table III - print the whole table in the same page.

Table iV - improve indentation. It is difficult to read.

There are also strange interrupted lines. 

Fig7. caption of Fig.7 is in the page following the figure. In addiction, it is not clear what Fig.7(a) represents: what does "population" mean?

Table 5 - Caption in wrong page. Improve indentation

subsection caption 4.1 in wrong page

subsection caption 4.3 in wrong page

SUGGESTION

In these vehicles the electrical motors that are used are not only synchronous PM motors, but other studies have been carried out where the adoption of synchronous wound-rotor motors are applied. I like to cite:

 C. Rossi, D. Casadei, A. Pilati, and M. Marano, “Wound rotor salient pole synchronous machine drive for electric traction,” in Conference Record of the 2006 IEEE Industry Applications Conference Forty-First IAS Annual Meeting, vol. 3, 2006, pp. 1235–1241.

(authors are the same of [72]. Other works are for tractors).

Other electric motors that allows interesting performance are the Hybrid excited PM motors. I suggest to cite such motors, e.g.:

 Y. Amara, S. Hlioui, H. B. Ahmed, and M. Gabsi, “Power capability of hybrid excited synchronous motors in variable speed drives applications,” IEEE Transactions on Magnetics, vol. 55, no. 8, pp. 1–12, 2019.

 D. Michieletto, L. Cinti, and N. Bianchi, “Hybrid excitation PM synchronous motors: Part II - finite element analysis,” IEEE Transactions on Energy Conversion, pp. 1–10, 2021.

and an interesting comparison between these motor type is in 

 L. Cinti, D. Michieletto, N. Bianchi, and M. Bertoluzzo, “A comparison between hybrid excitation and interior permanent magnet motors,” in 2021 IEEE Workshop on Electrical Machines Design, Control and Diagnosis (WEMDCD), 2021, pp. 10–15.

Finally, let me congratulate with the authors for their nice review.

Reviewer 3 Report

the paper is very interesting and I have read it with pleasure. There are many references to optimal papers.

The setting of the article is appreciable. The division of the paragraphs is also the result of the articles present in the literature.

What I would like to point out, to give a more scientific cut, is this: off-road electrification is not the result of the ecological goodness of the manufacturers, but is due to a binding constraint on the power classes. The power of the means is not present in this article, and it is a serious shortcoming. I suggest inserting a paragraph with the list of current regulations that have led to the development of hybrid technology. They have tier and stage name based on the state (American and European), there are limits 56 kW, 130 kW, a little discussion would be needed.

Round 2

Reviewer 2 Report

I see that the paper is improved.

It is a pity that the Authors do not include the suggested papers to improve the References.

I loose time to look for those for their paper. 

Author Response

Point 1: I see that the paper is improved.

Response 1: Thank you so much for your kind appreciation.

Point 2: It is a pity that the Authors do not include the suggested papers to improve the References.

I loose time to look for those for their paper.

Response 2: The Authors express their regret for not being able to include the suggested papers. As explained in our response in Round 1, the topics of those papers were beyond the scope of our work. Our paper reports the motors those were used in past commercial and demonstration products as well as research. It does not go into further details of motors. As none of the suggested papers are on off-road construction and agricultural equipment applications of electric motors, they are not included.

Reviewer 3 Report

thank for following my suggestions. Very good work

Author Response

Point 1: thank for following my suggestions. Very good work

Response 1: The Authors sincerely appreciate the Reviewer’s kind approval.